# A Review on the Usability, Flexibility, Affinity, and Affordability of Virtual Technology for Rehabilitation Training of Upper Limb Amputees

**DOI:** 10.3390/bioengineering10111301

**Published:** 2023-11-09

**Authors:** Xiangyu Liu, Di Zhang, Ke Miao, Yao Guo, Xinyu Jiang, Xi Zhang, Fumin Jia, Hao Tang, Chenyun Dai

**Affiliations:** 1College of Communication and Art Design, University of Shanghai for Science and Technology, Shanghai 200093, China; liuxiangyu@usst.edu.cn (X.L.); miaoke@usst.edu.cn (K.M.); 2Department of Geriatrics, Xinhua Hospital, Shanghai Jiao Tong University School of Medicine, Shanghai 200092, China; zhang-di@sjtu.edu.cn; 3School of Information Science and Technology, Fudan University, Shanghai 200433, China; yguo19@fudan.edu.cn; 4School of Informatics, The University of Edinburgh, Edinburgh EH8 9AB, UK; xinyu.jiang@ed.ac.uk; 5Department of Industrial Design, Hanyang University, Ansan 15586, Republic of Korea; xizhang@hanyang.ac.kr; 6Institute of Science and Technology for Brain-Inspired Intelligence, Fudan University, Shanghai 200433, China; 7School of Biomedical Engineering, Shanghai Jiao Tong University, Shanghai 200241, China; chenyundai@sjtu.edu.cn

**Keywords:** upper prosthetic rehabilitation, upper limb amputation therapy, virtual rehabilitation, computer-aided rehabilitation

## Abstract

(1) Background: Prosthetic rehabilitation is essential for upper limb amputees to regain their ability to work. However, the abandonment rate of prosthetics is higher than 50% due to the high cost of rehabilitation. Virtual technology shows potential for improving the availability and cost-effectiveness of prosthetic rehabilitation. This article systematically reviews the application of virtual technology for the prosthetic rehabilitation of upper limb amputees. (2) Methods: We followed PRISMA review guidance, STROBE, and CASP to evaluate the included articles. Finally, 17 articles were screened from 22,609 articles. (3) Results: This study reviews the possible benefits of using virtual technology from four aspects: usability, flexibility, psychological affinity, and long-term affordability. Three significant challenges are also discussed: realism, closed-loop control, and multi-modality integration. (4) Conclusions: Virtual technology allows for flexible and configurable control rehabilitation, both during hospital admissions and after discharge, at a relatively low cost. The technology shows promise in addressing the critical barrier of current prosthetic training issues, potentially improving the practical availability of prosthesis techniques for upper limb amputees.

## 1. Introduction

Upper limb amputation poses many physical and psychological challenges that can significantly constrain one’s daily activities and occupational abilities [1,2,3,4]. Fortunately, with the development of human–machine interface technology, neuroprostheses can provide hand-like control feelings to help amputees recover their hand function. Recent advances in prosthetic technology have significantly improved the functional capacity of prosthetic devices [5,6] via sophisticated human–machine systems that can be seamlessly integrated with the user’s nervous system. However, the latest investigations show that the abandonment rate of prostheses is still as high as 44% [7] due to poor compatibility between users and devices.

The abandonment of upper limb prostheses can be attributed to several factors. First, despite intelligent prosthetic limbs providing a natural interaction between the user and machine, there were no significant improvements in self-reported functional recovery [8]. Participants suggested that a longer adaptation time may be necessary for significant functional improvements [9,10,11,12,13,14,15], indicating that extended training may be required to gain a natural control of the prosthesis [16,17]. Second, participants expressed interest in performing specific activities that they had been unable to achieve, highlighting the need to address individual functional needs and goals in prosthetic design and training. Third, the functions provided by prosthetic limbs may not satisfy the requirements of users, especially for activities requiring coordinated individual digit control [18,19,20,21], such as buttoning shirts and tying shoes. It is noteworthy that these reasons for the abandonment of prostheses with natural control may vary across individuals, suggesting that a new solution is strongly desired to effectively address these challenges [22].

In contrast, computer-aided prosthetic training is a promising course of action for tackling the challenges associated with patient rehabilitation compliance, particularly those related to cost and motivation for training [23,24,25]. Combined with virtual reality technologies [26], human–machine interaction and multimodal sensing, computer-aided virtual prosthetic training can provide an interesting, virtual, and flexible training ecosystem for individuals with upper limb amputations [27,28]. Personalized and immersive experiences enable patients to simulate movements and undertake tasks that are consistent with their specific needs in a controlled environment [29]. In addition, computer-aided prosthetic training has a great potential to reduce the costs of money and time [30] over traditional prosthetic training [8], since this technology can allow patients to conveniently train at home without real prostheses [29,30].

Overall, although the potential for virtual prosthetic rehabilitation exists as a viable, versatile, and cost-effective tool for amputation therapy, the extent of its effectiveness remains unclear. A systematic review is necessary to assess how to integrate this technology into traditional rehabilitation techniques to optimize its benefits. To obtain a more profound understanding of the therapeutic value of virtual environments for upper limb amputees, this study offers a comprehensive review of this topic and presents a broad discussion of the feasibility of virtual technology to enhance prosthetic rehabilitation. Firstly, we provide a detailed analysis of the theoretical applications of virtual technology to prosthetic rehabilitation. Secondly, we present a summary of current trends and future challenges using this technique. Our survey also seeks to identify gaps between the current research and clinical applications and propose potential avenues for future investigations in this field. The goal of this review is to contribute to the existing body of research by providing the benefits of virtual technology in upper limb rehabilitation and completing numerous investigations into its potential applications.

## 2. Methods

The paper systematically evaluates a wide range of studies on prosthetic rehabilitation using computer-aided virtual technology. Given the plethora of available literature of many quantities and types, an approach with a more integrative and evidence-based perspective on upper limb amputation and associated prosthetic rehabilitation was used in this review. Accordingly, we adopted a five-stage framework for our review, as demonstrated by Whittemore and Knafl [31], consisting of the following steps: Problem Identification, Literature Search, Data Evaluation, Data Analysis, and Results Presentation [31]. We used Preferred Reporting Items for Systematic Reviews and Meta-Analyses (PRISMA) [32] to show how we selected and filtered the relevant articles (see review guideline in Figure 1). The PRISMA framework is a standard tool commonly used in the literature reviews of biomedical technology [33,34].

### 2.1. Purpose of the Review

This review aims to critically evaluate the direct empirical evidence concerning the utilization of virtual technology in the prosthetic control rehabilitation of upper limb amputees. Furthermore, this review seeks to comprehensively analyze various approaches to upper limb amputation rehabilitation. The synthesis of the findings from the study offers valuable insights into the potential benefits of utilizing virtual technologies in overcoming the challenges associated with upper limb amputation. By exploring these approaches, all stakeholders could enhance their comprehension and refine upper limb rehabilitation strategies. This review facilitates the presentation and analysis of studies conducted within this particular scope. Ultimately, this research provides essential knowledge and guidance for practitioners and patients, enabling them to navigate the complexities of prosthetic rehabilitation and improve occupational and daily life outcomes for upper limb amputees.

### 2.2. Literature Research

The study extensively searched electronic sources using various databases to identify relevant research studies (Table 1 depicts the search strategy and keyword term). The keywords included phrases such as “Upper limb amput*”, “Hand amput*”, “Upper extremity amput*”, “Prosthetic training”, “Rehabilitation”, “Prosthe*”, “Video game”, and “Virtual reality”, as well as combinations of relevant keywords in the databases. Specifically, virtual reality is defined as an interactive medium that immerses users in a computer-simulated environment and enables users to navigate and manipulate virtual environments, objects, and characters. To ensure consistency and accuracy, the research excluded any combinations including “Stroke”, “Parkinson’s”, and “Cerebral” to avoid other neurological disease-related interferences with the results. The literature was researched comprehensively using the following databases: PubMed, Cochrane Library, EMBASE, Web of Science, and Scopus, focusing on studies published between 1 January 2010 and 31 August 2023.

### 2.3. Inclusion and Exclusion Criteria

The inclusion criteria encompass research employing computer simulation environments as a means of presenting upper extremity amputation rehabilitation. Such simulations replicate real-life scenarios, offering a virtual platform for assessing the effectiveness of diverse rehabilitation methods. Thus, performing a thorough investigation of the subject matter, we have expanded the purview of the initial notion concerning computer-assisted stimulation by encompassing diverse domains such as three-dimensional visualization, game-based entertainment, augmented reality, etc.

In the first stage, the first author evaluated all articles and excluded unrelated and duplicated articles. Then, all papers were separately reviewed by the five researchers who are experienced in industrial design and biomedical engineering. The researchers reviewed all abstracts and keywords, and the other criteria of included articles were as follows:

The title, abstract, or keywords contain at least one of the following terms: upper limb amputation, upper extremity amputation, hand amputation, virtual reality, or video game;The study type is either a research paper, clinical trial, or review;The article is written in English.

The exclusion criteria were as follows:

Other types of research include conference papers, communication, keynotes, or book chapters;Poor scores were obtained after quality evaluation of the STROBE [35] and CASP [36] standards.

### 2.4. Data Analysis

The selected articles underwent assessment using the STROBE checklist for observational studies and the CASP tool for clinical research. Specifically, the STROBE checklist was utilized to evaluate the title, abstract, introduction, method, results, discussion, and other relevant information of the article, and the CASP tool was applied to assess the validity, methodology, performance, and applicability of the article.

In the selection of CASP and STROBE standards, we followed the following principles:The study details an original virtual rehabilitation system devised by the author, which stands out due to its unique approach that avoids the need for participant recruitment or simulated research. This innovative system, implemented according to the CASP guidelines, brings important advancements to the field.After analyzing the research involving the recruited participants, we found that the selection criteria demographic or epidemiological information of the selected individuals were not discussed. We used CASP for this analysis.The remaining aspects of the study adhere to the STROBE guidelines.

We have presented the criteria for evaluating the quality of literature within the fields of CASP and STROBE, which are shown in Appendix A, Table A2 and Appendix A, Table A3, respectively. The scoring criteria employed are as follows: a positive response (YES) is indicative of 1 point, whereas negative responses (NO) and non-applicable (NA) responses do not contribute towards the overall score. For a piece of literature to be deemed eligible for inclusion by CASP, it must attain a minimum score of 5 out of the 10-point scoring criteria. Similarly, for inclusion by STROBE, the literature must achieve a minimum score of 15 out of the 33-point scoring criteria.

The data analysis was accomplished through inductive content analysis, which entailed a thorough reading of the full texts to gain a comprehensive understanding of the contents. The resulting themes were identified based on their similarities and differences and then organized into distinct categories, as presented in Appendix A, Table A1. The first author conducted the review, and the three other authors were consulted afterward. The relevant findings and themes were discussed, and a consensus process was used to confirm the methodology and themes.

## 3. Results

After eliminating duplicate articles, a total of 34,146 articles were obtained. 20 were finally selected using the inclusion and exclusion criteria after a meticulous appraisal (Figure 1). All authors engaged in secondary research to ensure the most recent relevant articles were included. The selected 20 papers underwent a rigorous examination, encompassing various aspects such as subject information, research methodology, assessment, and outcomes. Moreover, the checklist scores and Critical Appraisal Skills Programme (CASP) scores for each study were also evaluated (see Appendix A). Here, we briefly summarized the main contents of the 20 selected works.

Simon and his colleagues conducted a study to investigate the benefits of virtual technologies for individuals with upper limb amputations [37]. Their findings indicate that virtual technologies offer a range of advantages in this context. These benefits include the ability to practice and refine control skills, real-time performance measurement, increased flexibility in therapy settings, enhanced safety, support for rehabilitation programs, and the facilitation of comparative analysis. The researchers specifically highlighted the effectiveness of the Target Achievement Control (TAC) test, a virtual evaluation tool, in measuring real-time performance. This assessment tool provides valuable control-related information, enabling the assessment and comparison of different control strategies. Furthermore, the TAC test allows for customization of the testing environment and captures essential data related to the effectiveness of control algorithms in the study. It is important to note, however, that the benefits and outcomes discussed in this paper are specific to the virtual technologies examined and may not encompass all potential advantages or discoveries in this field.

Rahman Davoodi [38] conducted a study that explored the advantages of using virtual technologies in the context of upper limb amputations. The study highlighted several benefits of virtual technologies, such as the ability to provide a realistic simulation, instant modification of scenarios, and cost-effective training. Recent studies have also demonstrated the successful use of electromyography (EMG) control in virtual environments, showing comparable efficacy to physical simulators for training EMG commands. Furthermore, real-time training in immersive virtual environments has been found to be effective. These quantitative findings emphasize the importance of incorporating realism in training programs to facilitate skill transfer. As a result, virtual technologies have the potential to offer a safe and engaging setting for the training and rehabilitation of individuals with upper limb amputations who use neural prostheses.

According to a study by Kaliki et al. [39], virtual technologies offer numerous advantages for upper limb amputees, such as restoring reaching abilities, improving hand control and orientation, non-invasive command schemes, adaptability, potential training with other subjects’ data, and quantifiable performance metrics. The study’s quantitative results demonstrate a significant increase in the percentage of trial completion for the ICS group, from 70% to 98% over ten sessions. Both the ICS group and the Natural Control group showed a decrease in the mean time taken to complete a trial, with notable differences between the groups. The sensitivity analysis identified limitations in achieving accuracy with the ICS, while spatial variability analysis revealed an overall improvement across the workspace. Overall, these findings provide valuable insights into enhancing performance, differentiating groups, and understanding the sensitivity of the ICS in upper limb prostheses.

Powell et al. [40] discuss the benefits of using virtual technologies in pattern recognition training for myoelectric prostheses. They present qualitative findings on the training and performance of a wrist disarticulation amputee using a pattern recognition training system to control a myoelectric prosthesis. The subject initially faced difficulties in executing various hand grasps and had overlapping movement classes. However, with training, the subject was able to replicate the muscle activity in the intact limb and generate movement with the virtual prosthesis. The training aimed to enhance the distinguishability of each movement and eliminate overlap. Eventually, the subject reported being able to discern and perceive the differences between each movement, considering them as feeling natural. The paper underscores the significance of consistency and distinguishability in muscle patterns for effective pattern recognition control. It emphasizes the need for practice, repetition, and slight adjustments to enhance the distinctiveness of each movement. Furthermore, the authors highlight the challenges of coordinating hand and wrist movements in pattern recognition control and stress the importance of accounting for the entire phantom hand and wrist position. In summary, the qualitative results emphasize the success of pattern recognition training in improving the control of a myoelectric prosthesis for individuals with wrist disarticulation amputation.

In a study by Bunderson [41], it was found that virtual environments offer flexibility and modifiability, enabling personalized rehabilitation programs and testing of control strategies. The lessons learned from virtual systems can also be applied to physical prostheses, improving their design and use. Interactive and realistic simulations in virtual environments provide valuable data for assessing control methods and real-time feedback. The study’s results show improvements in the time to completion, grip force, and distribution of the desired class signal. It was found that non-amputee subjects experienced a 29% improvement in completion time, while a shoulder disarticulation subject showed a 31% improvement. Grip force data indicated that non-amputee subjects exerted a higher average grip force compared to the average pinch force for humans. The most frequently used class by non-amputee subjects was Endpoint In, and a strong correlation was observed between the elbow flex/extension distance and completion time. These findings highlight the effectiveness of virtual technologies in enhancing task completion and understanding control strategies.

Blana et al. [42] also presented a study. Their use of virtual technologies for upper limb amputees’ rehabilitation offers several advantages and quantitative results. These include improved training and rehabilitation through realistic simulations, enhanced prosthesis control with advanced algorithms, data collection and analysis for assessing performance, personalized programs tailored to individual needs, real-time feedback for adjustments, increased engagement and motivation, and promising outcomes in terms of motor skills, range of motion, and prosthesis control. It is important to note that specific results may have varied depending on the particular virtual reality system and rehabilitation program employed. In the testing phase conducted online, participants successfully completed a target-reaching task with a path efficiency of 78% and minimal overshooting (1.5%). Furthermore, the accuracy of the artificial neural network (ANN) in predicting elbow flexion/extension and forearm pronation/supination angles exceeded the required targets, with a root mean squared error (RMSE) of 2.7 degrees and 5.5 degrees, respectively. These outcomes serve as evidence for the feasibility and effectiveness of utilizing combined kinematic and electromyographic signals from the proximal humerus to control virtual technologies.

Marek Kurzynski et al. [43] investigated the applications of virtual technologies for upper limb amputee rehabilitation. They found that these technologies offer advantages such as realistic simulations, improved prosthesis control, data analysis for performance evaluation, customization, feedback, engagement, and positive outcomes in motor skills, range of motion, and prosthesis control. Their research also focused on the differences in brain activity during a motor imagery task between a patient and a control subject. The patient showed a higher total power of the EEG, indicating the involvement of more brain regions. After motor imagery training, the patient’s total power decreased but remained higher than the control subject. The patient also had a higher proportion of lower frequency bands, suggesting an increased brain activation and attention to internal processing. Both the patient and control subjects had increased relative power of the alpha band after training, indicating a shift towards internally directed attention during the task.

N A Hashim et al. [44] conducted a study investigating the effects of virtual technologies on upper limb amputees’ muscle coordination and overall control, patient engagement and motivation, motor skills development in real environments, and overall patient outcomes when compared to traditional physiotherapy exercises. Their findings revealed remarkable improvements in motor coordination and muscle strength after the participants underwent the video game rehabilitation protocol. Notably, the virtual Box and Block Test (BBT) scores exhibited a consistent increase over the course of ten sessions, indicating a substantial enhancement in hand functionality and control. Additionally, the maximum voluntary contraction (MVC) values demonstrated a noticeable increase, suggesting a substantial improvement in muscle strength. Notably, the MVC values were found to be higher in able-bodied participants in comparison to trans-radial amputee participants, with the lowest values recorded in participants with congenital amputation. The modified Intrinsic Motivation Inventory (IMI) questionnaire and user evaluation survey revealed participants’ enjoyment and motivation towards the gaming experience, with positive ratings given for interest/enjoyment, perceived competence, perceived choice, pressure/tension, and value/usefulness. Consequently, the quantitative results strongly support the potential benefits of incorporating video games into the rehabilitation of upper limb amputees, as they enhance motor coordination, muscle strength, and patient motivation.

The effectiveness of game-based technologies in the rehabilitation of upper limb amputees has been explored by Prahm et al. [45]. They demonstrate that such interventions offer numerous advantages, including increased motivation and engagement, improved proficiency in prosthesis control, and the ability to train at home. In their study, game-based interventions were found to result in greater improvements in electromyographic (EMG) control compared to traditional methods. This was evident through patients’ enhanced ability to switch between degrees of freedom and activate muscle signals proportionally. Moreover, positive changes were observed in patients’ myoelectric aptitude, as well as short-term impacts on EMG control parameters. In addition, game-based interventions were found to increase patient motivation, performance, and effort during rehabilitation. These findings highlight the efficacy of game-based interventions in enhancing prosthesis control and promoting patient engagement in the rehabilitation process.

The importance of virtual technologies in prosthetic training for upper limb amputees has been emphasized by Perry et al. [46]. Virtual technologies offer several advantages, including enhanced engagement, real-time feedback, customization, and a safe training environment. These features have been shown to be effective in improving motor control accuracy, functional performance, and user satisfaction, as supported by quantitative results from the study. Consequently, the findings underscore the potential of virtual technologies to enhance prosthetic training and contribute to an improved quality of life for individuals with upper limb amputations.

Hargove’s study [47] highlighted that virtual technologies offer a range of advantages in training and rehabilitation, including cost-effectiveness, rapid development, powerful technological capabilities, and training specificity. The quantitative results from the study indicate that virtual technologies can lead to improvements in functional control and performance for upper limb amputees. Specifically, completion time for the TAC test, as well as performance in the SHAP and Box and Block Tests, significantly improved after a 6-week home trial (decreased from 7.5 to 5.5 s). Strong correlations were observed between virtual and physical outcome measures, suggesting the potential transferability of skills from virtual environments to real-life prosthesis control. However, further research is needed to validate these findings and extend the research to a larger population.

Winslow et al. [11] conducted a study examining the potential benefits of virtual technologies for improving rehabilitation success rates. By providing myositis training outside of the clinical environment, virtual technologies offer advantages over current training approaches. Moreover, they offer engaging and mobile training options and enable remote access to patient performance metrics. The study yielded quantitative results that demonstrated increased control of the wrist flexor and extensor muscles in the non-dominant limb. It also highlighted the potential for personalized training and identified the need for alternative EMG approaches. Concerns regarding transferability and time course were also raised. It is recommended that future research includes longitudinal studies to evaluate the impact of pre-prosthetic training methods on functional outcomes in amputees.

DT Kluger’s study [48] emphasizes the various advantages of virtual technologies in the realm of upper prosthetics. Firstly, virtual reality environments with virtual prosthetic hands present a more cost-effective option compared to physical prostheses. The upfront cost of implementing virtual technologies is significantly lower. Moreover, virtual technologies circumvent physical limitations by providing a means for testing and evaluation without necessitating the use of a physical prosthesis or socket fitting. This eradicates challenges such as maintenance, discomfort, and fatigue commonly associated with physical prostheses. Furthermore, virtual environments eliminate secondary sensory cues, ensuring a controlled and standardized testing environment. They also enable a high repeatability and standardization across different laboratories, thereby promoting reliable and comparable results. In addition, virtual technologies facilitate real-time analysis through the digital logging of every contact interaction, allowing for the precise examination of motor control and sensory output. Lastly, they afford enhanced control and sensory feedback, thus enabling investigations into closed-loop control behavior and the observation of behavioral changes with or without feedback.

Yoshimura’s study [49] provides an evaluation of the main applications associated with virtual technologies. The authors discuss how these technologies offer enhanced motor skill learning, increased motivation and engagement, personalized and adaptable training, real-time feedback and monitoring, as well as a safe and controlled environment for rehabilitation. The study also presents quantitative results that demonstrate a significant improvement in motor function and prosthetic control following virtual reality (VR)--based rehabilitation. Moreover, the findings indicate a positive correlation between higher levels of immersion and the acquisition of skills. These results underscore the effectiveness of virtual technologies in enhancing motor function and prosthetic control in individuals with upper limb amputations.

Brain Kaluf et al. [50] conducted a study to evaluate the advantages that virtual technologies provide for upper limb amputees. The authors found that virtual technologies offer several key benefits, including intuitive control, enhanced accuracy, personalized prosthetic design, potential for early intervention, and a non-invasive approach. Furthermore, the study’s quantitative analysis revealed that calibration accuracy was high and completion rates were similar between the sound limb and residual limb, regardless of age and previous prosthesis experience. These findings emphasize the efficacy and accessibility of virtual technologies in improving prosthetic control for individuals with upper limb amputations.

In their recent article, Garske et al. [10] examine the practical applications of virtual technologies in the context of upper limb amputees. Specifically, they investigate the potential use of game-based prosthetic training, rehabilitation, and motor learning. Utilizing a quantitative approach, the authors present the findings of their study, which indicate a positive attitude and enthusiasm towards game-based rehabilitation from the participants. However, the study also highlights several barriers, including usability, game design, training, and challenges that need to be addressed. Notably, the participants expressed expectations for muscle development, improved prosthetic ability, and additional benefits such as the alleviation of phantom limb pain. In order to facilitate these outcomes, the participants emphasized the importance of engaging gameplay, a shared experience among users, and the involvement of all stakeholders in the design process. Furthermore, challenges were identified in areas such as justifying the use of serious games, design issues, stakeholder involvement, and knowledge sharing. Overall, this study offers valuable insights into the expectations, preferences, and challenges associated with game-based prosthetic training for individuals with upper limb amputations.

R. Nataraj et al. [51] have provided an extensive analysis of the various applications of virtual technologies in control rehabilitation. The authors discuss the significant roles of control mode variations, performance measures, and perception measures in this domain. It is emphasized that control modes can be adjusted to optimize both performance and cognition. Performance measures play a crucial role in evaluating key factors such as reaching path length, movement accelerations, and joint torque output. The study also delves into the differential impact of control mode variations on the performance of dominant and non-dominant hands. This aspect holds importance in guiding the customization of virtual training for rehabilitation purposes based on individual needs and abilities.

Lucas EI Raghibi et al. [52] investigated the applications of virtual technologies in control training for upper limb amputees. Their study explored various uses, including the use of mini-games to familiarize amputees with electromyography (EMG) signal control, the testing of prostheses in a virtual reality environment, the objective evaluation of performance in virtual tasks, and the subjective evaluation of usability, through surveys. Their quantitative results revealed a significant correlation between task duration and subjective usability scores for each prosthesis. However, they found no correlation between the changes in the task duration and usability scores. The authors highlighted the variability in the learning curves and perceived usability among participants, which ultimately led to changes in their preferred devices after the virtual reality sessions.

Segas et al. [53] have recently investigated the potential of virtual reality (VR) training, hybrid arm control, and a generic model for prosthesis control in promoting the performance of reaching and grasping tasks by upper limb amputees. Encouraging results were observed in this study, with participants achieving high success rates using the hybrid arm control and reaching a median success rate of 93%. Additionally, the generic model for prosthesis control also yielded favorable outcomes. When compared to natural arm movements, only minor differences were observed in both success rates and movement times. To further validate the effectiveness of the virtual technologies, a physical proof of concept involving a teleoperated robotic platform was utilized. These findings collectively suggest that virtual technologies show considerable promise in training and enhancing the capabilities of upper limb amputees, equipping them with the ability to perform tasks traditionally executed by natural arm movements.

Hunt and colleagues [54] conducted a study involving upper limb amputees to assess the impact of virtual technologies on their functionality. The aim of the research was to investigate the effects of virtual reality (VR) and augmented reality (AR) training on each participant’s ability to use a physical prosthesis, as well as the potential for skill transference between virtual and physical reality tasks. The researchers also explored the influence of limb loading during virtual object interactions on the participants’ proficiency and examined the relationship between offline training accuracy and online functionality. The findings of this study have important implications for the use of virtual technologies in improving functionality and skill transfer for upper limb amputees.

Subsequently, a comprehensive analysis of the included articles was performed to determine their usability, flexibility, affinity, and affordability (Figure 2). The analysis was carried out meticulously to provide a comprehensive and complete perspective.

## 4. Discussion

For the virtual rehabilitation training of upper limb amputees, the goal is to help the amputees adapt to prostheses intuitively and as soon as possible so that they can return to normal life with the machine. However, the process of adapting to physical prostheses faces many difficulties due to the interactive design of the hardware and control strategy software. Accepting important feedback from users regarding the adaptation progress is problematic once the hardware has been constructed since any system revisions at this point may cost a lot of money or time. To overcome these difficulties, the virtual environment can simulate the upper limb prosthesis where the control strategy or hardware design can be easily tested and updated. Our findings are summarized with four characteristics of virtual prosthetic rehabilitation, including:

Usability: Virtual approaches provide an intuitive training environment to improve motor functions step-by-step and alleviate the ceiling effect of physical rehabilitation.

Flexibility: Virtual approaches provide an extendable and flexible platform for multi-technology interaction.

Affinity: Virtual approaches enhance adherence and motivation for users to persist in long-term training.

Affordability: Virtual environments offer a low-cost and adjustable platform to enhance the skill proficiency of upper limb amputees.

The above four key points, respectively, encompass the primary concerns regarding the prosthetic rehabilitation of upper limb amputatees: therapy efficacy, clinical applicability, rehabilitation motivation, and costs. Accordingly, this review is shows the great potential of virtual technology in solving these issues of prosthetic rehabilitation application.

### 4.1. Usability

Control training: The utilization of virtual environments constitutes a potential evolutionary platform that enhances usability for individuals with upper limb amputation in different control training tasks. The training focuses on enhancing the control ability of their prosthetic limb efficiently and effectively via simulating various activities of daily life using virtual technology, including grasping a cup, pressing buttons, shaking hands, etc. Importantly, virtual modalities provide an intuitive setting that fosters rehabilitation, overcoming the inherent challenges posed by physical limitations.

Recent studies have found that the control skills gained from the virtual environment can be transferred to physical training [38,43,44,46,48,55]. This integrated platform can adjust its parameters to adapt to the physical conditions of amputees and their achieved control abilities [10,37,48]. Accordingly, the usability of virtual technology can enable a gradual rehabilitation, which allows amputees to enhance their control skills smoothly and robustly [36], thus eliminating the ceiling effect common in physical recovery. For instance, Kaliki et al. [39] found that control performance, notably bottle reaching, catching, and moving a mouse, gradually improved with the adaptable settings in the virtual environment. In particular, game-based rehabilitation in virtual environments is effective as it can extend the training period and has a high probability of replayability [56]. Another recent work [40] demonstrated that a computer-based pattern recognition training system can be used to learn and improve pattern recognition skills. Game developers can integrate therapy-relevant tasks within the game context and tailor them to personal needs. Besides physical rehabilitation, the virtual environment can also provide augmented feedback to help amputees regain their sense of initiative. Moreover, mental and physical rehabilitation can also be achieved, especially in learning and neuroplasticity [57]. This non-motor training can indirectly benefit the control ability of the prosthesis.

Patient-centered: Virtual technology shifts the perspective from the technology to the patient [41,47,51], stimulating new prosthetic designs and recovery tasks [41]. To achieve improved control skills and muscle strength, amputees can undergo specific training programs in the virtual environment, which facilitates shorter recovery periods [45] and eliminates physical limitations [47]. Efforts within the academic community have been made to actively promote the concept of a patient-centered design. One example of such efforts is the study conducted by Prahm et al. [45] which explored the feasibility of utilizing gaming platforms for rehabilitation purposes. This study sought to evaluate various parameters, including the efficacy of electromyography (EMG) control before and after the intervention, as well as participants’ motivation, performance, and effort. Notably, the findings of this research exhibited an overall improvement in EMG control and fine muscle activation following the intervention. Moreover, patients reported experiencing higher levels of enjoyment when engaged in racing games, while rhythm-based games were deemed particularly advantageous for optimizing EMG control. Ultimately, their work concluded that incorporating game-based interventions can serve as a valuable adjunct to traditional training methods, thereby potentially yielding superior clinical outcomes and practicing the patient-centered design concept.

Recent developments have also shown encouraging possibilities for enhancing the adaptation process of prosthetic devices in upper limb amputees [58,59,60,61]. Since the functions of prostheses are predominantly interactive, the skills related to virtual prosthetic interactions can be leveraged to achieve physical realism [62]. Moreover, virtual prosthetic usage can provide a large amount of individual data, such as muscle activation, prosthetic torque, and generated force, which are critical for evaluating the control strategy and design efficacy of real prostheses [50,63]. Overall, virtual testing paradigms and complete tasks have been developed in a virtual environment to better assess [37,38,63,64] and refine prosthetic training techniques [41,42].

### 4.2. Flexibility

Co-creation: Virtual computer-aided solutions present a versatile co-creation platform for enhancing interdisciplinary collaboration since the field of prosthetics involves medicine, electronics, mechanics, computer science, etc. The digital platform for virtual upper limb prostheses is easily accessible for collaborators working together. This approach can potentially revolutionize various areas of research and training through its ability to synthesize data, analyze complex phenomena, and simulate real-world scenarios. For example, the integration of virtual reality and machine learning technologies, in particular, has facilitated the development of sophisticated control models that foster the interaction between the virtual and physical worlds. Blana. D et al. [42] highlighted that conducting experiments within an immersive virtual reality setting holds promise for gathering substantial and informative insights regarding the efficacy of controller performance. The current investigation accentuates the inherent prospects of employing sophisticated AI algorithms and signal processing techniques with the aim of augmenting the operational efficiency and user-friendliness of myoelectric prostheses. Although it remains challenging to integrate the multidisciplinary synergy of prosthetic training using a platform, this cooperation is now becoming the consensus in interdisciplinary research [10].

Engaging: Amputees can benefit from immersive experiences through the flexibility of virtual environments. Recent technological advances have led to the development of various simulated scenarios. These scenarios can include activities that may not be possible with traditional rehabilitation in a natural environment [63]. Furthermore, the virtual approach can also promote mental rehabilitation and neuroplasticity, encouraging a complete recovery [57]. Virtual systems can improve training interventions for motor and cognitive functions by creating interactive scenarios simulating real-world contexts with immediate feedback. In addition, a primary concern for amputees is the lack of adequate training time to enable neuroplastic adaptation. In contrast, the training time for virtual technology is relatively flexible due to the low requirements of the prosthetic devices.

A successful case for the use of a virtual reality simulator, named the Virtual Integration Environment (VIE), is examined by BN Perry et al. [46] as a training platform for individuals with upper extremity (UE) loss to acquire skills in controlling advanced prosthetic limbs. Thirteen active-duty military personnel with UE loss participated in passive motor training sessions utilizing the VIE. During these sessions, they mimicked the movements of a virtual avatar using their residual and phantom limbs. Surface electromyography (sEMG) from the residual limb was recorded to identify the movement intent of the users and then used as the control input of the avatar. Additionally, eight participants underwent active motor training sessions, during which they maneuvered a virtual avatar through various motion sets. The findings revealed that the VIE training platform effectively facilitated the training of individuals with UE loss in mastering advanced prosthetic control paradigms. Remarkably, the participants demonstrated the ability to generate different muscle contraction patterns in their residual limbs in terms of their movement intention, which can be accurately interpreted by pattern recognition algorithms embedded in the virtual platform. Overall, this study suggests that the VIE holds promise in rapidly and effectively training individuals with UE loss to operate advanced myoelectric prostheses by capitalizing on pattern recognition feedback or similar control systems.

However, to realize an engaging experience for the control of virtual prostheses is still a formidable challenge faced by individuals with limb loss. The most important issue is how to establish a two-way interaction between the user and the virtual environment. To address this, Rahman Davoodi and Gerald E. Loeb [38] proposed a physics-based target shooting game that aided amputees in mastering control of their prostheses. On the one hand, participants employed neural commands to manipulate the movements of a simulated prosthesis. On the other hand, participants also received a comprehensive range of feedback, including visual, auditory, and performance-based cues and haptics from the virtual system throughout the gameplay. The utilization of virtual training environments in this context showcases their potential benefits for amputee patients.

### 4.3. Affinity

Empathy: The stakeholders engaged in the rehabilitation of upper limb amputees possess an intimate familiarity with the distinct requirements of individuals in varying circumstances based onthe virtual environment, thereby enabling the tailoring of appropriate training and rehabilitation regimens. Through direct observation of the rehabilitation needs of upper limb amputees, these stakeholders are able to conceive customized interventions tailored to address the specific challenges encountered by each individual. This personalized approach amplifies the efficacy of these training and rehabilitation activities. Consequently, the term “empathy” in this paper denotes the stakeholders’ capacity to comprehensively grasp the rehabilitative expectations stemming from the hardships endured by upper limb amputees, thereby fostering an empathetic connection with their rehabilitation needs.

Hashet et al. [44] investigated the efficacy of video games in the rehabilitation of individuals with upper limb amputations. The study implemented a 4-week comprehensive rehabilitation program, which enrolled ten participants: five individuals with amputations and five able-bodied individuals for comparison. During the program, all participants demonstrated notable enhancements in both muscle strength and coordination. Moreover, a rigorous statistical analysis revealed a compelling and affirmative association between the duration of the training period and the scores obtained in the Box and Block Test. These findings provide compelling evidence in support of the hypothesis that video games possess the potential to augment motivation and enhance adherence amongst individuals undergoing upper limb amputee rehabilitation. R. Nataraj et al. [51] aimed to explore how limb dominance influences motor and perceptual behaviors in a virtual reality (VR) environment. Sixteen participants with healthy limb function were involved, each controlling a virtual hand using their own hand movements with various control adaptations. The results revealed a significant positive relationship between performance and binding for the dominant hand, which refers to the estimation of the time interval between a beep sound and grasp contact. Notably, there were intriguing variations in performance and binding depending on handedness and the specific control mode utilized. As a result, the researchers recommend that VR paradigm developers consider limb dominance when optimizing their settings to enhance performance and engagement. Moreover, tailoring VR rehabilitation techniques based on handedness may provide advantages for individuals with unilateral impairments, promoting greater empathy and inclusivity in rehabilitation practices. This enhanced realism allows for a more accurate representation of upper limb amputees’ physical and psychological experiences.

Adherence: Although various therapy techniques have been used, motivation remains a significant obstacle to the amount of training required for optimal recovery [56]. Virtual technology can be designed to increase motivation during prosthesis adaptation, as low motivation can result in abandonment. The integration of immersive and interactive elements can enhance the appeal and compliance of training programs, especially in populations with poor motivation or limited access to traditional prosthesis training. Moreover, subjective motivation provides a certain mental rehabilitation of neuroplasticity [42]. Powell, M.A. and Thakor, N.V. [40] demonstrated that each individual amputee approached pattern recognition with varying degrees of initial success, influenced by long-term motivation incentives. For example, Kurzynski et al. [43] explores the application of a computer-aided training (CAT) system to enhance the efficacy of preoperative mental training in individuals undergoing upper limb transplantation. Specifically, the authors propose an innovative system that utilizes virtual reality (VR) technology and sensory feedback to stimulate the sensorimotor cortex, thereby promoting the recovery of upper limb movement control. The CAT system comprises three key components: a three-dimensional virtual world generator, a sensory feedback mechanism that converts virtual hand-object interactions into mechanical vibrations, and a therapist’s panel for training supervision. Furthermore, the document elucidates the scientific rationale and motivation underlying this approach, including the capacity for cortical plasticity and reorganization through training-induced mechanisms. BD. Winslow et al. [11] presents a novel method of training myoelectric prosthesis users through the use of a mobile game-based training system. The study investigates the efficacy of this approach in enhancing key factors essential for successful myoelectric prosthesis usage. The findings indicate a notable advancement in these aspects among able-bodied participants who employed their non-dominant limb as a model for pre-prosthetic training in amputees. Additionally, participants reported high levels of usability and motivation when utilizing the game-based training approach. This study implies that mobile game-based myositis training has the potential to enhance rehabilitation success rates by offering training opportunities outside of the clinical setting. It concludes by recommending future research to assess the impact of pre-prosthetic training methods on prosthesis acceptance, wear time, functional outcomes, quality of life, and return to work.

Adaptability: Considering the potential adverse psychological effects of the virtual environment is also critical. Therefore, virtual approaches must be designed to enable adaptability depending on the mental states of users. Virtual technology can also modulate the difficulty, challenge, goal, feedback, interaction, and socialization of the training tasks to avoid possible frustration from amputees [56]. Recent work [43] demonstrated that controlling voluntary movements in transplanted hands remains a significant challenge in the field of upper limb transplantation. To address this obstacle, a promising approach involves the utilization of mental training techniques along with visual and sensory feedback, which have been demonstrated to facilitate structural and functional reorganization of the sensorimotor cortex. Initial findings from case studies using the computer-assisted training (CAT) system have shown encouraging results, as there have been positive changes in brain activity related to motor control. The advancement of the CAT system can be accomplished through the incorporation of techniques for biosignal classification and analysis. Prahm et al. conducted a study aimed at evaluating the impact of game-based interventions on prosthetic motor rehabilitation. The research included a sample of 14 patients with upper extremity amputation and 10 able-bodied participants. The participants were allocated into three distinct groups, namely the game-based intervention group, the control group, and the able-bodied control group. In the game-based intervention group, participants engaged in playing three different games that required muscle activation for control. The study assessed various parameters related to electromyographic (EMG) control both before and after the intervention, in addition to evaluating participant motivation, performance, and effort. Upon analyzing the findings, an improvement in overall EMG control and fine muscle activation was observed. Notably, patients reported that racing games were perceived as more enjoyable, while rhythm-based games were deemed better suited to provide challenges specifically related to EMG control. Consequently, the study concludes that the integration of game-based interventions serves as a valuable adjunct to standard EMG training, yielding enhanced clinical outcome measures.

### 4.4. Affordability

Budget constraints often lead to a lack of prioritization for new technologies, especially if current solutions are already beneficial. In a survey focused on physician attitudes toward new healthcare technology, it was found that 80% of respondents identified a lack of financial support as the main barrier [65]. However, patient experience also plays a significant role in technology adoption. When users have a more positive experience, it leads to increased engagement and treatment adherence, which adds value to the technology and helps justify the implementation costs [66]. Thus, the value that a solution provides relative to the cost of its implementation is a crucial factor in determining its adoption. Previous investigations demonstrated that more than half of upper extremity amputees face significant occupational obstacles, such as income reduction or even profession change [3,4,67]. Financial difficulties are a primary concern of amputees because inducing neuroplastic adaptation may be a long-term process, and they often cannot afford the high costs. Fortunately, existing evidence has demonstrated that the usage of a virtual environment can be affordable for upper limb amputees [10].

Low-cost: Budgetary constraints often lead to a decreased importance placed on innovative technologies, particularly when existing solutions are deemed sufficient in delivering benefits [65]. The virtual environment offers a niche benefit for the small population of upper limb amputees as it is financially affordable and widely compatible [10,46,47,49,50]. After hospital discharge, amputees require long-term and personalized rehabilitation support [1,5]. A virtual platform is low-cost as the amputee can be confined to a controllable platform which is suitable for in-home use [68]. In addition, the hardware design and control strategy can be easily adjusted for individuals to achieve the best adaptation between the prosthesis and the user in the virtual environment at a low cost. The optimal adaptation in the virtual environment can provide an essential reference for the configuration of real prostheses.

Virtual technology offers numerous cost-saving benefits for upper limb amputees. First, unlike traditional prostheses, whose hardware items require frequent adjustment or replacement, the adaptability of virtual technology allows for software customization, eliminating the costs of physical components. Second, virtual technology can integrate realistic and haptic simulations to closely replicate the functionality of a natural limb. This immersive experience prompts more effective training with less rehabilitation time, potentially reducing the associated costs. Third, virtual training allows for remote monitoring and telerehabilitation, where healthcare professionals can track progress, offer guidance, and make necessary adjustments remotely, reducing the need for in-person visits and saving travel expenses. In conclusion, the adaptability of hardware, the ability to mimic natural limb functioning, and the potential for remote monitoring and telerehabilitation make virtual technology a more cost-effective solution for upper limb amputees compared to traditional prosthetic items, resulting in significant long-term cost savings.

We conducted a cost-effectiveness analysis for the benefits of implementing virtual technology in motor rehabilitation compared with traditional methods. Specifically, we sought to determine if virtual approaches are more economically advantageous for all parties involved. We focused on stroke patients who also need motor function rehabilitation due to a lack of available data on prosthesis rehabilitation. From the patient’s perspective, traditional physical therapy services incur an annual cost of USD 11,689 per patient [69]. In contrast, utilizing an in-clinic virtual rehabilitation service involved a one-time payment of USD 1490, significantly lower than the cost of traditional therapy. Virtual telehealth services could further reduce costs to as low as USD 835 [70]. These findings suggest that virtual rehabilitation may offer a more cost-effective solution for patients. Lower treatment costs are important for both patients and insurance companies, as they increase the likelihood of insurance coverage, endorsement, and patient acceptance [30].

Free software: Virtual technology with open-source software can be leveraged to offer a free solution for crafting virtual environments. Prominent examples of such software are Blender and Unity [13,71], which have gained widespread usage in the customization of virtual environments. A notable attribute of these environments is their focus on catering to the needs of amputees. Specifically, upper extremity amputees may benefit from using such virtual environments to perform exercises designed to replicate daily life activities. For instance, Blana et al. [42] conducted a study wherein an artificial neural network was cultivated and subsequently assessed offline in a virtual reality setting based on open-access software. Their findings addressed the inherent constraints of myoelectric prostheses by devising a more intuitive and organic control mechanism. Utilizing these technologies, the scope for augmenting rehabilitative care becomes extensive. The intensity level of therapy can be easily adjusted without any extra costs according to the user’s needs.

NE Bunderson [41] discusses the use of an interactive virtual dynamic environment for testing control strategies for neural machines integrating with artificial limbs. The virtual environment is low-cost, easily configurable, and offers a wealth of data for post hoc analysis compared with real physical prostheses and robots. The document focuses on the development of a physics engine called Neuromechanic, which is used to simulate the dynamics of a virtual limb. The command extraction process converts raw EMG signals to the desired class command signals, which are then used by the control module to generate command torques. The document also presents the results of a test involving four non-amputee subjects and one shoulder disarticulation subject, who were able to successfully transfer blocks in the virtual environment at an average rate of just under two blocks per minute. Overall, the document highlights the potential of interactive virtual environments for testing and improving control strategies for prosthetic devices. Their findings show the successful transfer of blocks in the virtual environment by non-amputee and shoulder disarticulation subjects.

By tailoring the intensity of the virtual workouts to the patient’s specific capabilities and progress, therapists can potentially leverage these environments to contribute positively to the patient’s rehabilitation progress. Consequently, creating virtual environments using open-source software presents a promising avenue for advancing rehabilitative care for this population.

### 4.5. Future Challenges and Directions

In this study, we analyze factors that can contribute to increased patient engagement and improved adherence to prosthetic rehabilitation training through virtual technology. These factors form the basis of practical benefits for virtual therapy in amputee rehabilitation. Our process draws on the following four characteristics of the virtual environment: usability for prosthesis training, flexibility for the training platform, an affinity for user experience, and affordability for training costs.

Notably, the factors extracted from our findings can potentially inform the future design and development of virtual systems for rehabilitation training. However, several challenges must be addressed to fully realize the potential of virtual technology in amputee rehabilitation (Figure 3). These include the need for further investigation into the optimal interface design and sensory feedback for close-loop therapy platforms, and the establishment of rigorous protocols for assessing the efficacy of virtual rehabilitation interventions.

Challenges:

Advances in virtual prosthetic training-based prosthesis rehabilitation are predominantly limited to academic research settings, with few practical applications in clinical environments.We identified a need for more research focused on incorporating closed-loop control of user feedback (especially prosthetic tactile sensation, such as material, texture, temperature, etc.) into immersive virtual environments.One significant challenge to the widespread adoption of the technology discussed in this article lies in the limited number of amputee subjects involved in the experimental trials. Such trials pose challenges in obtaining generalizable results. As scientific knowledge advances, it becomes increasingly crucial to incorporate user-centered designs and larger sample sizes in studies related to clinical deployments. Subsequently, it becomes necessary to evaluate whether remote training options can support these studies.The elderly population may encounter challenges when using virtual rehabilitation technology. First, they may have deficiencies in the fundamental skills required for operating smart electronic devices. Second, the deterioration of neuromuscular function may result in the inability to operate the virtual human–machine system.

Directions:

Improving realism: This can be achieved by integrating novel technologies (e.g., haptic feedback-increased degrees of freedom in virtual environments) and using realistic visual cues in simulated tasks. In addition, there is a need to enhance closed-loop control to improve the accuracy of motion tracking and achieve more natural movements in virtual settings. Improvement of virtual reality environments and virtual limbs in rehabilitation paradigms has been proposed to enhance their efficacy. The modeling of virtual limbs with a higher degree of realism and movement resolution has the potential to yield substantial benefits. In addition, future research should evaluate the impact of pre-prosthetic training methods on various outcomes in amputee populations.Enhancing closed-loop control: Enhanced closed-loop control refers to optimizing the communication between the user and the device, allowing for more natural and intuitive movements. This can be achieved through advanced sensors, machine learning algorithms, and real-time feedback. By improving the accuracy and response speed of the device, patients may experience greater control and improved function.Integrating multiple modalities: Furthermore, there is a growing need to explore the use of various modalities in virtual rehabilitation. Advances in this direction can result in the development of personalized multimodal interventions tailored to individual needs and preferences. Such interventions can integrate visual, auditory, and haptic feedback to create rich and immersive training experiences.Supporting reintegration: Reintegration support involves providing patients with the necessary resources and support to successfully reintegrate into society following amputation and rehabilitation. This may include vocational training, peer support groups, and mental health services. By addressing the psychosocial aspects of rehabilitation, patients may enjoy more fulfilling lives.Persistent attention has been devoted to serious games within the realm of rehabilitative practices over an extended period. Our study delves into the utilization of game-based approaches in the context of prosthetic rehabilitation with a specific emphasis on upper limb training. To ascertain the effectiveness of games in upper limb prosthetic rehabilitation, it is imperative to prioritize extensive, long-term experiments conducted within individuals’ homes, as they yield substantial evidence of their efficacy.Virtual environment paradigms should consider limb dominance to optimize their settings for better performance and perceptional engagement. Adapting rehabilitation for handedness may benefit unilateral impairments [51].

Addressing these challenges will be critical to successfully translating virtual rehabilitation technologies into clinical practice, where they could significantly contribute to the goal of improving healthcare delivery for amputees. Moreover, the implementation of virtual prosthetic training in the field of prosthesis rehabilitation has primarily been restricted to academic research, resulting in a dearth of practical applications in clinical settings. This phenomenon is readily noticeable as there exists a gap in the translation of research outcomes to real-world scenarios. In addition, the efficacy of virtual technology in long-term rehabilitation following upper limb amputation calls for additional empirical evidence. Although virtual technology has shown significant benefits, establishing a standard protocol for virtual technology-based rehabilitation training is still very important to clinical use. Nevertheless, it can be foreseen that this technology will have great application prospects in various stages of the rehabilitation of upper limb amputees. The lack of a widely accepted framework for virtual prosthetic training could be attributed to the current limitations in data collection, data processing, and user interfaces. Implementing adequate protocols for the aforementioned challenges can ensure a smooth transfer of virtual prosthetic training techniques from academic research to clinical practice.

## 5. Conclusions

As discussed in this review, we have drawn the following conclusions:Virtual environments are effective for prosthetic rehabilitation in amputees as they can accomplish control training through virtual environments and transfer their control experience to the real prostheses.The technological components outlined in this study, such as signal control, sensation feedback, adaptability, and gamification, can be combined for the purpose of enhancing user engagement and facilitating autonomous learning within and beyond clinical settings.The integration of immersive serious gaming and visual and somatosensory feedback using virtual technology, in particular, holds immense potential for promoting prosthetic embodiment within the virtual space.

Virtual technology provides a flexible and configurable platform for upper limb amputees in their control rehabilitation in both admissions to the hospital and discharge. The virtual environment integrates multidisciplinary approaches associated with the timely and latest technologies from the fields of biomedical engineering and computer science. Long-term rehabilitation after discharge can also benefit from virtual restoration in terms of prosthetic adaptation, occupational rehabilitation, dignity recovery, and the reintegration of psychosocial status.

## Figures and Tables

**Figure 1 bioengineering-10-01301-f001:**
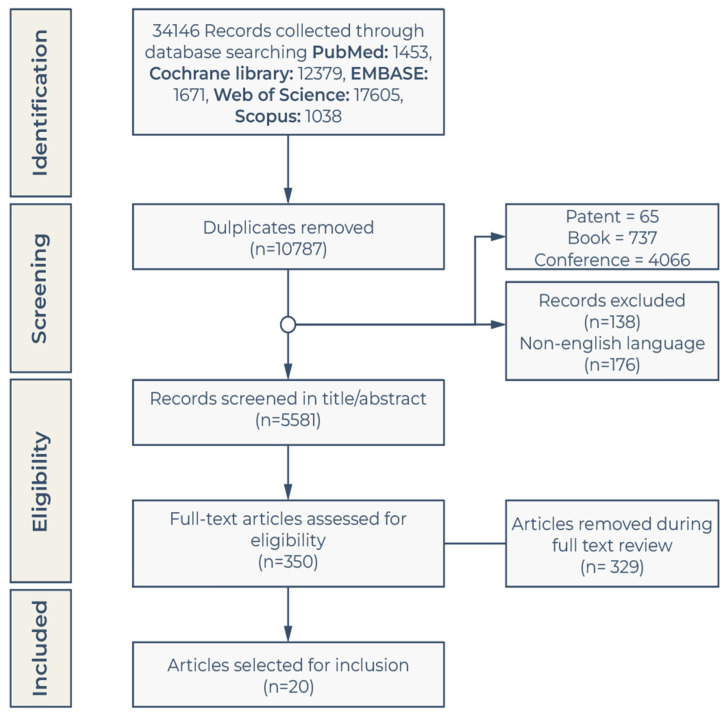
PRISMA diagram.

**Figure 2 bioengineering-10-01301-f002:**
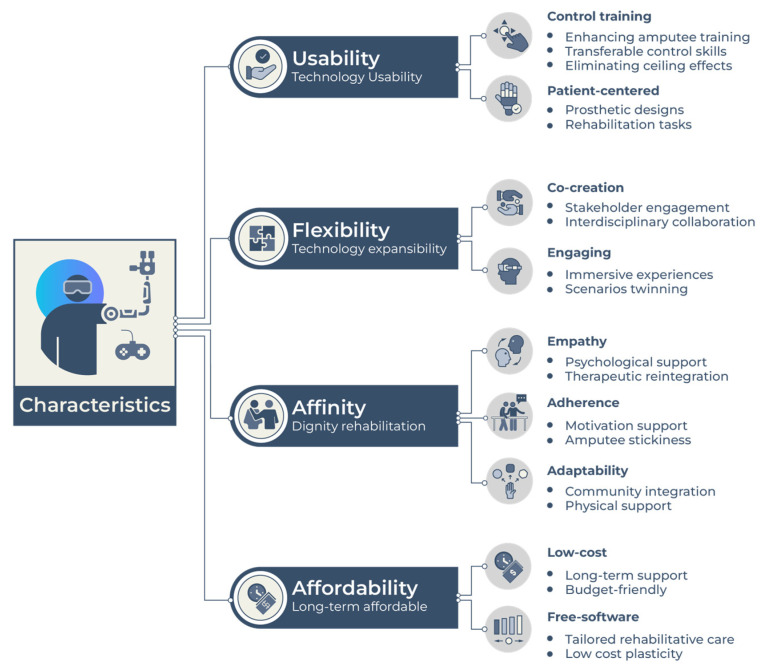
Characteristics of a computer-aided virtual approach for upper limb amputation rehabilitation.

**Figure 3 bioengineering-10-01301-f003:**
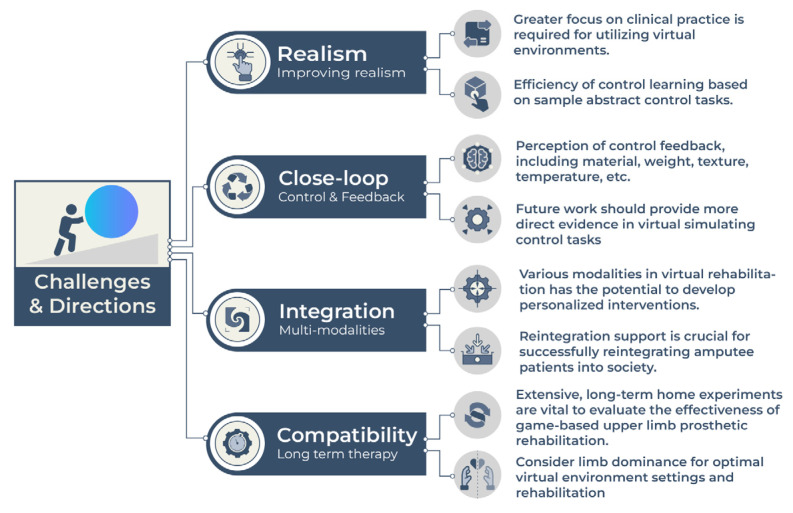
Future challenges and directions of virtual technology for upper limb amputees.

**Table 1 bioengineering-10-01301-t001:** Searching strategies in each database.

Strategy	PubMed	Cochrane Library	EMBASE	Web of Science	Scopus
((Upper limb amput*)) OR ((Hand amput*)) OR ((Upper extremity amput*)) AND ((Prosthetic training)) OR ((Rehabilitation)) AND ((Control)) AND ((Virtual reality)) NOT ((Stroke)) NOT ((Parkinson)) NOT ((Cerebral palsy))	859	843 trials	1209	9113 (6757 articles, 1016 reviews, 302 trials)	679
((Upper limb amput*)) OR ((Hand amput*)) OR ((Upper extremity amput*)) AND ((Prosthetic training)) OR ((Rehabilitation)) AND ((Control)) AND ((Video game)) NOT ((Stroke)) NOT ((Parkinson)) NOT ((Cerebral palsy))	542	638 trials	418	7927 (6014 articles, 845 reviews, 265 trials)	91
((Reintegration to normal live)) OR ((Reintegration to normal living)) AND ((Upper)) AND ((Amputation)) NOT ((Stroke)) NOT ((Parkinson)) NOT ((Cerebral palsy))	3	51 trials	1	160 (139 articles, 7 reviews, 7 trials)	23
((Reintegration to occupation)) OR ((Reintegration to work)) AND ((Upper)) AND ((Amputation)) NOT ((Stroke)) NOT ((Parkinson)) NOT ((Cerebral palsy))	7	61 trials	9	122 (106 articles, 11 reviews, 4 trials)	95
((Reintegration to daily life)) OR ((Self-care)) AND ((Upper)) AND ((Amputation)) NOT ((Stroke)) NOT ((Parkinson)) NOT ((Cerebral palsy))	42	10,786 trials	34	283 (241 articles, 34 reviews, 7 trials)	150
Total	1453	12,379	1671	17,605	1038

## Data Availability

The data presented in this study are openly available in PubMed, Cochrane Library, EMBASE, Web of Science, and Scopus.

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
