# Peer review of "A Review on the Usability, Flexibility, Affinity, and Affordability of Virtual Technology for Rehabilitation Training of Upper Limb Amputees"

_bioengineering, 2023, doi:10.3390/bioengineering10111301_

Round 1

Reviewer 1 Report

I recommend publication in its current form, as it is well written, with a clear topic and objectives, and follows the PRISMA guidelines. However, we would like to make a few minor corrections. First, the definition of VIRTUAL REALITY needs to be provided in more detail in INCLUSION CRITERIA. Second, we conducted a quality assessment of the individual studies included in this study, but the scores for the quality assessment are not presented. Only the names of the quality assessment are presented in TABLE 1, but please add the scores as well.

Author Response

Reviewer #1:

I recommend publication in its current form, as it is well written, with a clear topic and objectives, and follows the PRISMA guidelines. However, we would like to make a few minor corrections.

Response: Thanks for your comments. We feel great thanks for your professional review work on our article.

Comment R.1.1

First, the definition of VIRTUAL REALITY needs to be provided in more detail in INCLUSION CRITERIA.

Response: Thanks for your comments. In our manuscript, we followed the initial definition of Virtual Reality (VR), which is an interactive medium that immerses users in a computer-simulated environment and enables users to navigate and manipulate virtual environments, objects, and characters. In addition, to ensure a comprehensive examination of the topic, we have broadened the scope of the original concept within computer-aided stimulation, including 3D visualization, game-based entertainment, augmented reality, etc.

We have revised the content for the definition of Virtual Reality (VR) to be more appropriate, as copied below. (Highlighted in Lines 115-121, in “2.2 Literature research” Section)

The keywords included phrases such as "Upper limb amput*," "Hand amput*," "Upper extremity amput*," "Prosthetic training," "Rehabilitation," "Prothe*," "Video game," and "Virtual reality," as well as combinations of relevant keywords in the databases. Specifically, the virtual reality is defined as an interactive medium that immerses users in a computer-simulated environment and enables users to navigate and manipulate virtual environments, objects, and characters.

We have also revised the content for the definition of computer-aided stimulation to be more appropriate, as copied below. (Highlighted in Lines 130-136, in “2.3 Inclusion and exclusion criteria” Section)

The inclusion criteria encompass research employing computer simulation environments as a means of presentation for upper extremity amputation rehabilitation. Such simulations replicate real-life scenarios, offering a virtual platform for assessing the effectiveness of diverse rehabilitation methods. Thus, facilitating a thorough investigation of the subject matter, we have expanded the purview of the initial notion concerning computer-assisted stimulation, encompassing diverse domains such as three-dimensional visualization, game-based entertainment, augmented reality, etc.

Comment R.1.2

Second, we conducted a quality assessment of the individual studies included in this study, but the scores for the quality assessment are not presented. Only the names of the quality assessment are presented in TABLE 1, but please add the scores as well.

Response: Thanks for your comments. We presented the 10 criteria for assessing the quality of literature in CSAP and the 33 criteria for assessing the quality of literature in STROBE in Appendix B and Appendix C, respectively. The scoring criteria are as follows: YES receives 1 point, while NO and NA do not receive any points. To be eligible for inclusion, literature must obtain a score of 5 or higher among the 10 quality criteria in CASP. Similarly, to be eligible for inclusion, literature must obtain a score of 15 or higher among the 33 questions in STROBE.

We have revised the content to be more appropriate, as copied below. (Highlighted in Lines 166-173, in “2.4 Data analysis” Section):

We have presented the criteria for evaluating the quality of literature within the fields of CSAP and STROBE, which are shown in Appendix B and Appendix C, respectively. The scoring criteria employed are as follows: a positive response (YES) is indicative of 1 point, whereas negative responses (NO) and non-applicable (NA) responses do not contribute towards the overall score. For a piece of literature to be deemed eligible for inclusion by CASP, it must attain a minimum score of 5 out of the 10-point scoring criteria. Similarly, for inclusion by STROBE, the literature must achieve a minimum score of 15 out of the 33-point scoring criteria.

Reviewer 2 Report

Thanks for the opportunity to review the manuscript titled, " A Mini Review on the Usability, Flexibility, Affinity, and Affordability of Virtual Technology for Rehabilitation Training of Upper Limb Amputees".

This study conducted a systematic review to explore the advantages of utilizing Virtual technology in the prosthetic rehabilitation training of upper limb amputees.

The results summarised 4 possible benefits as well as 3 challenges in reality, affirming the promising future of integrating virtual technology into upper limb amputees. My major concerns are listed below.

1.         The authors employ a specific framework to evaluate the articles, as illustrated in Figure 2, but the origin of this framework is unclear to me. Is this framework a standard tool commonly used in the evaluation of rehabilitation technology, or was it specifically developed by the authors for this study?

2.         In the introduction, please explicitly write down the review's objective.

3.         I anticipated a review of the current state of virtual technology utilization for rehabilitation training in upper limb amputees, along with an analysis of papers based on the proposed framework. However, the presentation of the results and the ensuing discussions appear too general, often lacking specificity to the rehabilitation of UL amputees. The authors should provide a more detailed description that explicitly outlines what has been achieved using this technology with people who have UL amputations.

Below are specific comments for consideration:

4.         Title: I prefer to remove the word ‘Mini’

5.         Lines 61 meaning of ‘the issue’ is unclear

6.         In Lines 72-73, the authors mentioned the background and the aim of this review, but it seems vague. “Assess how to integrate the virtual technology into traditional rehabilitation” is not the objective of this review.

7.         In lines 73-76, the authors mentioned the therapeutic value of this technology, but in Appendix A of the included studies, the treatment effects were not shown, which poorly supports the results of the better outcome of virtual technology.

8.         Please remove Line 82 – 90. You do not need to outline the content of the manuscript in the introduction.

9.         The results part is too brief. Please describe the major finding in the results section.

10.      In lines 172-174, please describe in detail why the adaptation of hardware in virtual technology is more cost-effective compared to the traditional items.

11.      Line 187 and figure 2: What is the meaning of ‘control training’? this term seems very vague to me

12.      Line 204 should be physical rehabilitation instead of physiological.

13.      Line 220 ‘flexibility’ Any attempt of co-creation for people with UL amputation. The author cited a list of papers, but it remains unclear to me which one is related to UL amputation.

14.      Line 231 ‘Engaging’ the authors describe the advantage of virtual technology, that is to provide immersive experiences. But there is no evidence to show that this advantage is observed in the rehabilitation of people with UL amputation.

15.      In line 245-257, is there any meaning of “This immersive experience has triggered empathy for able-bodied participants” in this paragraph? Or only this sentence express the title “Empathy”?

16.      Line 270 ‘Affordability’ . The authors assert that the technology is financially affordable, yet they fail to provide any numerical data or specific figures from the study to substantiate this claim.

17.      Given the diverse types of virtual technology and environments, how does one evaluate the baseline functional level and design a personalized training plan? Additionally, how is the mid-term assessment conducted to adjust and tailor the plan as needed? If this is not currently achievable, please include a discussion on these limitations within your content.

18.      The subjects of the included articles tend to be younger people. For the popularization of virtual technology, are there different challenges when facing the older-age groups? Are there any differences in treatment effect?

Author Response

Reviewer #2:

This study conducted a systematic review to explore the advantages of utilizing Virtual technology in the prosthetic rehabilitation training of upper limb amputees.

The results summarized 4 possible benefits as well as 3 challenges in reality, affirming the promising future of integrating virtual technology into upper limb amputees.

Response: Thank you for providing us with your valuable expertise and insightful assessment of our article. We greatly appreciate your comments. 

Comment R.2.1

My major concerns are listed below. The authors employ a specific framework to evaluate the articles, as illustrated in Figure 2, but the origin of this framework is unclear to me. Is this framework a standard tool commonly used in the evaluation of rehabilitation technology, or was it specifically developed by the authors for this study?

Response: In this review, we employed the Preferred Reporting Items for Systematic Reviews and Meta-Analyses (PRISMA) framework [1] as a comprehensive guideline, which is a standard tool commonly used in the evaluation of biomedical technology [2,3]. Over the years, in tandem with the advancement of knowledge synthesis methods, various extensions to the original PRISMA Statement have been introduced [4–7]. The PRISMA guideline is widely recognized and extensively employed by researchers engaged in conducting systematic reviews and meta-analyses [8–10]. This guideline provides a structured framework that delineates the methods, outcomes, and elucidations of such studies, to ensure meticulousness and comprehensiveness in reporting. We illustrated the entire process of executing PRISMA in Figure 1. 

Reference:

  1. Moher, D.; Liberati, A.; Tetzlaff, J.; Altman, D.G. Preferred Reporting Items for Systematic Reviews and Meta-Analyses: The PRISMA Statement. Journal of Clinical Epidemiology2009, 62, 1006–1012, doi:10.1016/j.jclinepi.2009.06.005.
  2. Page, M.J.; Moher, D. Evaluations of the Uptake and Impact of the Preferred Reporting Items for Systematic Reviews and Meta-Analyses (PRISMA) Statement and Extensions: A Scoping Review. Syst Rev2017, 6, 263, doi:10.1186/s13643-017-0663-8.
  3. Caulley, L.; Cheng, W.; Catalá-López, F.; Whelan, J.; Khoury, M.; Ferraro, J.; Husereau, D.; Altman, D.G.; Moher, D. Citation Impact Was Highly Variable for Reporting Guidelines of Health Research: A Citation Analysis. Journal of Clinical Epidemiology2020, 127, 96–104, doi:10.1016/j.jclinepi.2020.07.013.
  4. Rethlefsen, M.L.; Kirtley, S.; Waffenschmidt, S.; Ayala, A.P.; Moher, D.; Page, M.J.; Koffel, J.B.; PRISMA-S Group; Blunt, H.; Brigham, T.; et al. PRISMA-S: An Extension to the PRISMA Statement for Reporting Literature Searches in Systematic Reviews. Syst Rev2021, 10, 39, doi:10.1186/s13643-020-01542-z.
  5. PRISMA-P Group; Moher, D.; Shamseer, L.; Clarke, M.; Ghersi, D.; Liberati, A.; Petticrew, M.; Shekelle, P.; Stewart, L.A. Preferred Reporting Items for Systematic Review and Meta-Analysis Protocols (PRISMA-P) 2015 Statement. Syst Rev2015, 4, 1, doi:10.1186/2046-4053-4-1.
  6. Tricco, A.C.; Lillie, E.; Zarin, W.; O’Brien, K.K.; Colquhoun, H.; Levac, D.; Moher, D.; Peters, M.D.J.; Horsley, T.; Weeks, L.; et al. PRISMA Extension for Scoping Reviews (PRISMA-ScR): Checklist and Explanation. Ann Intern Med2018, 169, 467–473, doi:10.7326/M18-0850.
  7. Hutton, B.; Salanti, G.; Caldwell, D.M.; Chaimani, A.; Schmid, C.H.; Cameron, C.; Ioannidis, J.P.A.; Straus, S.; Thorlund, K.; Jansen, J.P.; et al. The PRISMA Extension Statement for Reporting of Systematic Reviews Incorporating Network Meta-Analyses of Health Care Interventions: Checklist and Explanations. Ann Intern Med2015, 162, 777–784, doi:10.7326/M14-2385.
  8. Martucci, A.; Gursesli, M.C.; Duradoni, M.; Guazzini, A. Overviewing Gaming Motivation and Its Associated Psychological and Sociodemographic Variables: A PRISMA Systematic Review. Human Behavior and Emerging Technologies2023, 2023.
  9. Cacciamani, G.E.; Chu, T.N.; Sanford, D.I.; Abreu, A.; Duddalwar, V.; Oberai, A.; Kuo, C.-C.J.; Liu, X.; Denniston, A.K.; Vasey, B. PRISMA AI Reporting Guidelines for Systematic Reviews and Meta-Analyses on AI in Healthcare. Nature Medicine2023, 29, 14–15.
  10. Aslam, W.; Jawaid, S.T. Systematic Review of Green Banking Adoption: Following PRISMA Protocols. IIM Kozhikode Society & Management Review2023, 12, 213–233.

We have revised the content to be more appropriate, as copied below. (Highlighted in Lines 93-97, in “2. Methods” Section):

We used Preferred Reporting Items for Systematic Reviews and Meta-Analyses (PRISMA) [32] to show how we selected and filtered the relevant articles (see review guideline in Figure. 1). The PRISMA framework is a standard tool commonly used in the literature review of biomedical technology [33,34].

Comment R.2.2

In the introduction, please explicitly write down the review's objective.

Response: Thanks for your comments. We have elaborated on the purpose of this review in the introduction part to enhance the overall clarity and scientific rigor of our work.

We have revised the content to be more appropriate, as copied below. (Highlighted in Lines 72-85, in “1. Introduction” Section)

Overall, although the potential for virtual prosthetic rehabilitation exists as a viable, versatile, and cost-effective tool for amputation therapy, the extent of its effective-ness remains unclear. A systematic review is necessary to assess how to integrate this technology into traditional rehabilitation techniques to optimize its benefits. To obtain a more profound understanding of the therapeutic value of virtual environments for up-per amputees, this study offers a comprehensive review of this topic that presents a broad discussion of the feasibility of virtual technology to enhance prosthetic rehabilitation. Firstly, we provide a detailed analysis of the theoretical applications of virtual technology to prosthetic rehabilitation. Secondly, we present a summary of current trends and future challenges using this technique. Our survey also seeks to identify gaps between the current research and clinical applications and propose potential avenues for future investigations in this field. The goal of this review is to contribute to the existing body of research by providing the benefits of virtual technology in upper limb rehabilitation and complementing numerous investigations into the potential applications.

Comment R.2.3

I anticipated a review of the current state of virtual technology utilization for rehabilitation training in upper limb amputees, along with an analysis of papers based on the proposed framework. However, the presentation of the results and the ensuing discussions appear too general, often lacking specificity to the rehabilitation of UL amputees. The authors should provide a more detailed description that explicitly outlines what has been achieved using this technology with people who have UL amputations.

Response: Thanks for your comments. We have made several improvements to the content of our article to enhance clarity and precision. We extended the timeframe of our literature survey to include publications up until August 2023. For those selected papers, we provided a very detailed description in the Results section and an outline description in Table Appendix A. In addition, we added a new column, “Joints of upper-limb” to indicate which parts of upper-limb amputation the article focuses on. in Appendix A. This allows for a comprehensive analysis of the current state of research.

We have revised the content to be more appropriate, as copied below. (Highlighted in in “3. Results” Section, “4. Discussion” Section and Table Appendix A.

Below are specific comments for consideration:

Response: We greatly appreciate the feedback provided, as it enables us to improve the quality of our work. We have meticulously addressed each point raised and have documented as below.

Comment R.2.4

Title: I prefer to remove the word ‘Mini’.

Response: Thanks for your comments. We have removed the word ‘Mini’ from the title.

We have revised the title to be more appropriate, as copied below.

A Review on the Usability, Flexibility, Affinity, and Affordability of Virtual Technology for Rehabilitation Training of Upper Limb Amputees

Comment R.2.5

Lines 61 meaning of ‘the issue’ is unclear.

Response: Thanks for your comments. ‘The issue’ refers the concerns and the obstacles encountered during the upper extremity rehabilitation, specifically in the financial burden and the motivation required for their training. We have revised description about ‘the issue’ to ensure preciseness and eliminate any potential ambiguity in the manuscript.

We have revised the content to be more appropriate, as copied below. (Highlighted in Lines 61-63, in “1. Introduction” Section)

In contrast, computer-aided prosthetic training is a promising option to tackle the challenges associated with patient rehabilitation compliance, particularly those related to cost and motivation for training.

Comment R.2.6

In Lines 72-73, the authors mentioned the background and the aim of this review, but it seems vague. “Assess how to integrate the virtual technology into traditional rehabilitation” is not the objective of this review.

Response: Thanks for your comments. In the updated manuscript, we have incorporated a comprehensive analysis of the efficacy of virtual technology in the rehabilitation of individuals with upper limb amputations. In addition, we have revised the content for the objective of this review in the introduction.

We have revised the content to be more appropriate, as copied below. (Highlighted in Lines 72-85, in “1. Introduction” Section)

Overall, although the potential for virtual prosthetic rehabilitation exists as a via-ble, versatile, and cost-effective tool for amputation therapy, the extent of its effective-ness remains unclear. A systematic review is necessary to assess how to integrate this technology into traditional rehabilitation techniques to optimize its benefits. To obtain a more profound understanding of the therapeutic value of virtual environments for up-per amputees, this study offers a comprehensive review of this topic that presents a broad discussion of the feasibility of virtual technology to enhance prosthetic rehabilitation. Firstly, we provide a detailed analysis of the theoretical applications of virtual technology to prosthetic rehabilitation. Secondly, we present a summary of current trends and future challenges using this technique. Our survey also seeks to identify gaps between the current research and clinical applications and propose potential avenues for future investigations in this field. The goal of this review is to contribute to the existing body of research by providing the benefits of virtual technology in upper limb rehabilitation and complementing numerous investigations into the potential applications.

Comment R.2.7

In lines 73-76, the authors mentioned the therapeutic value of this technology, but in Appendix A of the included studies, the treatment effects were not shown, which poorly supports the results of the better outcome of virtual technology.

Response: Thanks for your comments. We have added a new column “Treatment effects” to show the therapeutic value of the technology in Appendix A.

We have revised the appendix A to be more appropriate, as copied below. (Highlighted in “ Appendix A” part)

Comment R.2.8

Please remove Line 82 – 90. You do not need to outline the content of the manuscript in the introduction.

Response: Thanks for your comments. We have removed the outline the content of the manuscript in the introduction and revised the introduction section.

Comment R.2.9

The results part is too brief. Please describe the major finding in the results section.

Response: Thanks for your comments. For those selected papers, we provided a very detailed description in the Results section and an outline description in Table Appendix A. This allows for a comprehensive analysis of the current state of research.

We have revised the content to be more appropriate. (Highlighted in Lines 190-433, in “3. Results” Section)

Comment R.2.10

In lines 172-174, please describe in detail why the adaptation of hardware in virtual technology is more cost-effective compared to the traditional items.

Response: Thanks for your comments.

One of the primary factors contributing to the cost-effectiveness of virtual technology for upper amputees is the adaptability of its hardware. Traditional prosthetic items typically necessitate frequent adjustments, modifications, or even complete replacement as the user’s needs evolve. In contrast, virtual technology offers greater flexibility in catering to the unique requirements of individuals through software customization. This adaptability substantially diminishes the need for additional physical components or costly replacements, thereby yielding long-term cost savings.

Moreover, virtual technology facilitates the utilization of hardware that closely replicates the functionality of a natural limb. Through the integration of virtual reality simulations and haptic devices, individuals can engage in interactive training sessions and receive real-world feedback. This immersive experience allows for more effective training with less rehabilitation time, potentially reducing the associated costs.

Another aspect of virtual technology that contributes to cost savings is its capacity for remote monitoring and tele-rehabilitation. Amputees can electronically connect with healthcare professionals who can remotely monitor their progress, offer guidance, and make necessary adjustments or programming changes. This obviates the requirement for frequent in-person visits, thereby saving both time and travel expenses for the patient.

In conclusion, the adaptability of hardware, the ability to mimic natural limb functioning, and the potential for remote monitoring and tele-rehabilitation make virtual technology a more cost-effective solution for upper amputees when compared to traditional prosthetic items. These advantages notably attenuate the need for frequent adjustments, replacements, and in-person visits, consequently leading to substantial long-term cost savings for individuals with upper limb amputations.

We have revised the content to be more appropriate, as copied below. (Highlighted in Lines 685-697, in “4.4. Affordability” Section, “Low-cost” part)

Virtual technology offers numerous cost-saving benefits for upper amputees. First, unlike traditional prosthesis whose hardware items require frequent adjustment or replacement, the adaptability of virtual technology allows for software customization, eliminating the costs of physical components. Second, virtual technology can integrate realistic and haptic simulations to closely replicate the functionality of a natural limb. This immersive experience prompts a more effective training with less rehabilitation time, potentially reducing the associated costs. Third, the virtual training allows for remote monitoring and tele-rehabilitation, where healthcare professionals can track progress, offer guidance and make necessary adjustments remotely, reducing the need for in-person visits and saving travel expenses. In conclusion, the adaptability of hardware, the ability to mimic natural limb functioning, and the potential for remote monitoring and tele-rehabilitation make virtual technology a more cost-effective solution for upper amputees compared to traditional prosthetic items, resulting in significant long-term cost savings.

Comment R.2.11

Line 187 and figure 2: What is the meaning of ‘control training’? this term seems very vague to me.

Response: Thanks for your comments. In our manuscript, the term ‘control training’ for upper amputees refers to a training regime designed to improve the functional control and coordination of the residual limb and any prosthetic device that may be used. This training focuses on enhancing the user’s ability to manipulate and utilize their prosthetic limb efficiently and effectively in various activities of daily living. The specific goals of control training may vary depending on the individual and their functional needs. Nevertheless, the overarching objective is to enhance the user’s ability to perform tasks with their prosthetic limb with greater dexterity, accuracy, and naturalness. The training may involve exercises that focus on strength, movement patterns, proprioception, sensory feedback integration, and neuroplasticity. The aim is to optimize the user’s ability to regain and maximize their functional independence, thereby improving their overall quality of life.

We have revised the content to be more appropriate, as copied below. (Highlighted in Lines 458-463, in “4.1. Usability”Section, “Control training”part.)

Control training: The utilization of virtual environments constitutes a potential evolutionary platform that enhances usability for individuals with upper-limb amputation in different control training tasks. The training focuses on enhancing the control ability of their prosthetic limb efficiently and effectively via simulating various activities of daily living using virtual technology, including grasping a cup, pressing bottoms, shaking hand, etc.

Comment R.2.12

Line 204 should be physical rehabilitation instead of physiological.

Response: Thanks for your comments. We have revised typographical error in the content.

We have revised the content to be more appropriate, as copied below. (Highlighted in Lines 479-481, in “4.1. Usability”Section, “Control training”part.)

Moreover, mental and physical rehabilitation can also be achieved, especially in learning and neuroplasticity.

Comment R.2.13

Line 220 ‘flexibility’ Any attempt of co-creation for people with UL amputation. The author cited a list of papers, but it remains unclear to me which one is related to UL amputation.

Response: Thanks for your comments. In our manuscript, Co-creation refers to the process of involving end-users or stakeholders in the design and development of new products or services. In the context of upper limb amputation, co-creation efforts can aim to create customized and personalized solutions that meet the specific needs and preferences of individuals with amputations. This can include the design of prosthetic devices, rehabilitation programs, and assistive technologies. All the papers cited are related to UL amputation. We have rephrased the statement to emphasize the papers are all about UL amputation.

We have revised the content to be more appropriate, as copied below. (Highlighted in Lines 511-512, in “4.2.Flexibility”Section, “Co-creation”part.)

The digital platform for virtual upper-limb prosthesis is easily accessible for collaborators to work together.

In addition, we take a specific example to support our statement. The work integrates machine learning technologies into virtual reality for the development of sophisticated control models that foster the interaction between the virtual and physical worlds.

We have revised the content to be more appropriate, as copied below. (Highlighted in Lines 517-522, in “4.2.Flexibility”Section, “Co-creation”part.)

Blana. D et al. [42] highlighted conducting experiments within an immersive virtual reality setting holds promise for gathering substantial and informative insights regarding the efficacy of controller performance. The current investigation accentuates the inherent prospects of employing sophisticated AI algorithms and signal processing techniques with the aim of augmenting the operational efficiency and user-friendliness of myoelectric prostheses.

Comment R.2.14

Line 231 ‘Engaging’ the authors describe the advantage of virtual technology, that is to provide immersive experiences. But there is no evidence to show that this advantage is observed in the rehabilitation of people with UL amputation.

Response: Thanks for your comments. We have added several examples to support the statement of “engaging” in “4.2. Flexibility” Section, “Engaging” part.

We have revised the content to be more appropriate, as copied below. (Highlighted in Lines 535-561 in “3.Results”Section.)

A successful case for the use of a virtual reality simulator named the Virtual Inte-gration Environment (VIE) is examined by BN Perry et al. [46] as a training platform for individuals with upper extremity (UE) loss to acquire skills in controlling advanced prosthetic limbs. Thirteen active-duty military personnel with UE loss participated in passive motor training sessions utilizing the VIE. During these sessions, they mimicked the movements of a virtual avatar using their residual and phantom limbs. Surface electromyography (sEMG) from the residual limb was recorded to identify the move-ment intent of the users and then used as the control input of the avatar. Additionally, eight participants underwent active motor training sessions, during which they ma-neuvered a virtual avatar through various motion sets. The findings revealed that the VIE training platform effectively facilitated the training of individuals with UE loss in mastering advanced prosthetic control paradigms. Remarkably, the participants demonstrated the ability to generate different muscle contraction patterns in their re-sidual limbs in terms of their movement intention, which can be accurately interpreted by pattern recognition algorithms embedded in the virtual platform. Overall, this study suggests that the VIE holds promise in rapidly and effectively training individuals with UE loss to operate advanced myoelectric prostheses by capitalizing on pattern recognition feedback or similar control systems.

However, to realize an engaging experience for the control of virtual prostheses is still a formidable challenge faced by individuals with limb loss. The most important is-sue is how to establish a two-way interaction between the user and the virtual environment. To address this, Rahman Davoodi and Gerald E. Loeb [38] proposed a physics-based target shooting game that aided amputees in mastering the control of their prostheses. On the one hand, participants employed neural commands to manipulate the movements of a simulated prosthesis.  On the other hand, participants also received a comprehensive range of feedback, including visual, auditory, and performance-based cues and haptics from the virtual system throughout gameplay. The utilization of virtual training environments in this context showcases their potential benefits for amputee patients.

Comment R.2.15

In line 245-257, is there any meaning of “This immersive experience has triggered empathy for able-bodied participants” in this paragraph? Or only this sentence expresses the title “Empathy”?

Response: Thank you for your comment. We apologize for any confusion caused by unclear descriptions. What we are trying to express here is that stakeholders involved in the rehabilitation of upper limb amputees have firsthand knowledge of the specific needs of individuals in different situations. This helps in customizing appropriate training and rehabilitation programs. By directly observing the rehabilitation needs of upper limb amputees, these stakeholders can design tailored interventions to address individual challenges. This personalized approach enhances the effectiveness of training and rehabilitation. Therefore, in the context of the article, “empathy” refers to the related stakeholders being able to have a clear understanding of the rehabilitation expectations based on the scenarios faced by upper limb amputees, thereby being able to empathize with their rehabilitation needs.

We have also revised the content to be more appropriate, as copied below. (Highlighted in Lines 563-573, in “4.3. Affinity”Section. “Empathy”Part.)

Empathy: The stakeholders engaged in the rehabilitation of upper limb amputees possess intimate familiarity with the distinct requirements of individuals in varying circumstances based on a virtual environment, thereby enabling the tailoring of ap-propriate training and rehabilitation regimens. Through direct observation of the rehabilitation needs of upper limb amputees, these stakeholders are able to conceive customized interventions tailored to address the special challenges encountered by each individual. This personalized approach amplifies the efficacy of training and rehabilitation activities. Consequently, the term “empathy” in this paper denotes the stakeholders’ capacity to comprehensively grasp the rehabilitative expectations stemming from the hardships endured by upper limb amputees, thereby fostering an empathetic connection with their rehabilitation needs.

Comment R.2.16

Line 270 ‘Affordability’. The authors assert that the technology is financially affordable, yet they fail to provide any numerical data or specific figures from the study to substantiate this claim.

Response: Thanks for your comments. In the updated manuscript, we have added a discussion on the concept of “affordability” for upper amputees. In this study, we have undertaken a comprehensive cost-effectiveness analysis to compare the merits of incorporating virtual technology in motor rehabilitation compared to traditional methodologies. Our primary objective was to elucidate whether the adoption of virtual approaches provides greater economic advantages for all stakeholders involved. Given the scarcity of data related to prosthetic rehabilitation, we analyzed a comprehensive dataset pertaining to stroke patients, aiming to establish and articulate the financial advantages associated with the implementation of virtual technology. 

We have also revised the content to be more appropriate, as copied below. (Highlighted in Lines 698-709, in “4.4. Affordability”Section. “Low-cost”Part.)

We conducted a cost-effectiveness analysis for the benefits of implementing virtual technology in motor rehabilitation compared with traditional methods. Specifically, we sought to determine if virtual approaches are more economically advantageous for all parties involved. We focused on stroke patients who also need motor function rehabilitation due to a lack of available data on prosthesis rehabilitation. From the patient’s perspective, traditional physical therapy services incur an annual cost of US$11,689 per patient [78]. In contrast, utilizing an in-clinic virtual rehabilitation service involved a one-time payment of US$1,490, significantly lower than the cost of traditional therapy. Virtual telehealth services could further reduce costs to as low as US$835 [79]. These findings suggest that virtual rehabilitation may offer a more cost-effective solution for patients. Lower treatment costs are important for both patients and insurance companies, as they increase the likelihood of insurance coverage, endorsement, and patient acceptance [80].

We also discussed the cost advantage of open-source software and added relevant arguments to the text.

We have also revised the content to be more appropriate, as copied below. (Highlighted in Lines 710-722, in “4.4. Affordability”Section. “Free software”Part.)

Virtual technology with open-source software can be leveraged to offer a free solution for crafting virtual environments. Prominent examples of such software are Blender and Unity [13,81], which have gained widespread usage in the customization of virtual environments. A notable attribute of these environments is their focus on catering to the needs of amputees. Specifically, upper extremity amputees may benefit from using such virtual environments to perform exercises designed to replicate daily life activities. For instance, Blana et al. [42]. Conducted a study wherein an artificial neural network was cultivated and subsequently assessed offline in a virtual reality setting based on open-access software. Their findings addressed the inherent constraints of myoelectric prostheses by devising a more intuitive and organic control mechanism. Utilizing these technologies, the scope for augmenting rehabilitative care becomes extensive. The intensity level of therapy can be easily adjusted without any extra costs according to the user’s needs.

Comment R.2.17

Given the diverse types of virtual technology and environments, how does one evaluate the baseline functional level and design a personalized training plan? Additionally, how is the mid-term assessment conducted to adjust and tailor the plan as needed? If this is not currently achievable, please include a discussion on these limitations within your content.

Response: Thanks for your comments. Since the use of virtual technology for amputation rehabilitation assist is a new solution in the research field, no literature has provided a standard rehabilitation protocol. However, we still think the requirement of establishing such a protocol for virtual-technology-based rehabilitation training is very important, including designing a personalized training plan, assessing rehabilitation efficacy and adjusting rehabilitation plan. However, we have to acknowledge that it is a formidable task for amputees to attain a comparable level of prosthesis usage as their intact limb previously, resulting in substantial variations in the duration and complexity of training across individuals. Thus, we tried to discuss the question proposed by the review for related future work. Instead of defining a precise protocol for the whole rehabilitation process, we have summarized the contributions of virtual technology in three main different rehabilitation stages before the use of a real prosthesis, as listed below:

  1. Pre-prosthetic Training: The objective of pre-prosthetic training is to adequately prepare the client and their limb for a well-fitting and fully functional prosthesis. This phase commences upon the closure of the wound and concludes with the attainment of a preparatory prosthesis. The duration of this phase varies depending on factors such as changes in limb volume, sensitivity, range of motion, physical condition of the residual limb, and the client’s psychological state. Virtual technologies can offer significant advantages in pre-prosthetic training by simulating various activities and exercises that help prepare the client for using a prosthesis. Virtual reality programs can provide interactive experiences that mimic real-life situations, allowing clients to practice limb movements, balance, and coordination.In addition, psychological support assumes great importance during the subacute phase as the client transitions emotionally from a state of “combat” survival to the realization that they will be living with an altered physicality for the remainder of their life. Psychological support entails imparting knowledge regarding the extensive possibilities of functionality with a prosthesis, providing an accurate understanding of prosthetic function, and introducing peer support or peer visitor programs to facilitate coping. Virtual reality can contribute to psychological support by creating immersive environments that help clients adjust to their altered physicality. This technology can provide virtual rehabilitation programs that educate clients about the functionalities of prostheses and offer psychological counseling sessions within the virtual environment.
  2. 2. Postural Exercises and Strengthening: Postural exercises are introduced to foster bodily symmetry and enhance upper quadrant strength. Clients engage in physical therapy programs aimed at improving flexibility, strength, and aerobic fitness of the trunk and lower body. Emphasis is placed on educating the client about altered body patterns and preventing incorrect postures that may give rise to overuse injuries. Postural exercises are introduced to foster bodily symmetry and enhance upper quadrant strength. Clients engage in physical therapy programs aimed at improving flexibility, strength, and aerobic fitness of the trunk and lower body. Emphasis is placed on educating the client about altered body patterns and preventing incorrect postures that may give rise to overuse injuries. Postural exercises and strengthening: Virtual technologies, such as motion-capture systems, can enable precise tracking of body movements during postural exercises and strengthening activities. By providing real-time feedback on posture and performance, virtual technologies can help clients maintain proper body alignment and prevent incorrect postures that may lead to overuse injuries.
  3. 3. Activities of Daily Living (ADL) Training: As wounds continue to heal, clients progress to more intricate ADLs encompassing tasks such as showering, dressing, and meal preparation. Compensatory techniques and adaptive equipment are introduced to promote independence. In some cases, retraining in hand dominance may be necessary, and clients are encouraged to master one-handed ADLs to ensure self-sufficiency during periods when the prosthesis may not be accessible. Virtual reality simulations, particularlycombined withEMG as the human-machine interface, can be used to train clients in performing tasks related to activities of daily living. These simulations can offer a safe and controlled environment for clients to practice using adaptive equipment and mastering one-handed ADLs. 

We have also revised the content to be more appropriate, as copied below. (Highlighted in Lines 813-826, in “4.5. Future Challenges and Directions”Section.)

Moreover, the implementation of virtual prosthetic training in the field of prosthesis rehabilitation has primarily been restricted to academic research, resulting in a dearth of practical applications in clinical settings. This phenomenon is readily noticeable as there exists a gap in the translation of research outcomes to real-world scenarios. In addition, the efficacy of virtual technology in long-term rehabilitation following upper limb amputation calls for additional empirical evidence. Although virtual technology has shown significant benefits, establishing a standard protocol for virtual-technology-based rehabilitation training is still very important to clinical use. Nevertheless, it can be foreseen that this technology will have great application prospects in various stages of rehabilitation for upper limb amputees. The lack of a widely accepted framework for virtual prosthetic training could be attributed to the current limitations in data collection, data processing, and user interfaces. Implementing adequate protocols for the aforementioned challenges can ensure a smooth transfer of virtual prosthetic training techniques from academic research to clinical practice.

Comment R.2.18

The subjects of the included articles tend to be younger people. For the popularization of virtual technology, are there different challenges when facing the older-age groups? Are there any differences in treatment effect?

Response: Thanks for your comments. Elderly individuals encounter challenges in upper limb amputation rehabilitation and exhibit deficiencies in fundamental skills required for operating smart electronic devices. This situation is exacerbated by the prolonged duration of the amputation process, which often leads to the development of single-handed habits detrimental to successful rehabilitation outcomes. Furthermore, the deterioration of neuromuscular function necessitates the reconstruction of muscle functionality for the restoration of neuromuscular control. Therefore, the older age group may not be suitable using the virtual training system. 

We have also revised the content to be more appropriate, as copied below. (Highlighted in Lines 771-774, in “4.5. Future Challenges and Directions”Section.)

Elderly population may encounter challenges when using the virtual rehabilitation technology. First, they may have deficiencies in fundamental skills required for operating smart electronic devices. Second, the deterioration of neuromuscular function may result in the inability to operate the virtual human-machine system.

Reviewer 3 Report

English level is fine.

Author Response

Reviewer #3:

The paper is a systematic review of the role of Virtual Technologies in the Rehabilitation process of upper limb amputees, to define tools and techniques to improve the acceptance rate and reduce the abandonment rate of prostheses.

The paper is well written and the full selection process of the papers which were included in the review is exceptionally documented and justified at each step.

After the selection of the final set of articles included in the review the four specific elements targeted by the review are explained in detail: Usability, Flexibility, Affinity, and Affordability.

We would like to express our deepest gratitude for your invaluable expertise and thorough evaluation of our article. Your insightful comments have greatly contributed to the enhancement of our work. We sincerely appreciate the time and effort you have dedicated to reviewing our manuscript.

Comment R.3.1

While the entire selection process is extremely well documented the other parts of the review are quite short. The introduction even though well documented it should introduce more detail in the problems which lead to the abandonment. This should be also related to the prosthesis itself, not just the rehabilitation. I would recommend here more detailed explanation of the problems themselves with more recent publications, which would correlate the specific problems with the current level of technology in prosthetics.

Response: Thanks for your comments. We have conducted an extensive literature review on the phenomenon of prosthetic abandonment, with a specific focus on the desertion of upper limb prostheses. In this article, we present our findings and present a revised introduction section that provides a comprehensive overview of the topic. Furthermore, we have made substantial revisions via providing detailed contents of reviewed papers to enhance the conclusions.

We have also revised the content to be more appropriate, as copied below. (Highlighted in Lines 48-60, in “1. Introduction”Section.)

The abandonment of upper limb prostheses can be attributed to several factors. First, despite intelligent prosthetic limb provides a natural interaction between the user and machine there were no significant improvement in self-reported functional recovery [8]. Participants recommended longer adaptation time may be necessary for significant functional improvements [9–15], indicating that extended training may be re-quired for the natural control of the prosthesis [16,17]. Second, participants expressed interest in performing specific activities that they had been unable to achieve high-lighting the need to address individual functional needs and goals in prosthetic design and training. Third, the functions provided prosthetic limb may not satisfy the requirements of users, especially for the activities requiring coordinated individual digit control [18–21], such as buttoning shirts and tying shoes. It is noteworthy that these reasons for prosthetic abandonment with natural control may vary across individuals, suggesting a novel solution is strongly desired to effectively address these challenges [22].

Comment R.3.2

While generally some of the high-quality papers will provide valid information as the get older, in a field where progress is made almost monthly having a lot of papers more than 7 years old it is expected that the virtual technologies have changed a lot and the reported data may not apply anymore to the current population. Even though the 17 selected articles are from 2011 and on, other parts of the reference list are quite older, and some might report data which is not valid anymore for the targeted points within the review. Also, I would suggest some detailing in the selection process of the 17 papers.

Response: Thanks for your comments. We have rigorously revised our references, encompassing literature up until August 2023, to ensure the utmost currency and relevance of the research. Additionally, we have meticulously explicated the inclusion criteria for the literature and meticulously annexed them in the Appendix B and C, thus furnishing readers with comprehensive transparency regarding our selection process.

We have revised the appendix A to be more appropriate, as copied below. (Highlighted in “Appendix. A” part)

Comment R.3.3

Searching PubMed there are many recent publications which deal with Virtual environments combined with real environments (or simply put augmented environments) which seem to provide also a lot of benefits to the patients.

Response: Thanks for your comments. We have updated our practice standards for incorporating literature, now extending the deadline until August 31, 2023, and conducting a new round of literature retrieval to encompass more recent sources. Furthermore, we have thoroughly revised the research details of all the included literature to ensure accuracy and relevance.

Comment R.3.4

The publication itself is quite short (6 pages with 2 presenting only the selection criteria for the paper) and while the conclusions are quite general, and I did get almost similar results using an AI bot and dropping a question regarding the advantages of Virtual technologies for upper limb prosthesis. So, in my opinion some data should be illustrated in a more detailed way to add value for the reader (namely to provide more information than a 1 minute interaction with an “advanced” search tool).

Response: Thanks for your comments. We would like to sincerely apologize for the lack of clarity in the initial manuscript. In response to your comment, we have made significant improvements to our revised manuscript. First, we have added a detailed description for the selected papers in the results section. Second, we have included supportive evidence that showcases the efficacy of virtual technology in enhancing the capabilities of individuals with upper limb amputations in the discussion section. This addition greatly strengthens our argument and provides researchers with robust evidence to support further investigations into this field.

Round 2

Reviewer 2 Report

The quality of the manuscript has been improved after the revision. The authors have misinterpreted my first comment.

My major concerns are listed below. The authors employ a specific framework to evaluate the articles, as illustrated in Figure 2, but the origin of this framework is unclear to me. Is this framework a standard tool commonly used in the evaluation of rehabilitation technology, or was it specifically developed by the authors for this study?

The framework I am referring to is the one listed in Figure 2, ie Usability, flexibility, Affinity and Affordability and the sub-characteristics under these major domains. I would like to know the origin and validity of this framework.

Author Response

Comment R.2.1

The authors employ a specific framework to evaluate the articles, as illustrated in Figure 2, but the origin of this framework is unclear to me. Is this framework a standard tool commonly used in the evaluation of rehabilitation technology, or was it specifically developed by the authors for this study?

The framework I am referring to is the one listed in Figure 2, ie Usability, Flexibility, Affinity and Affordability and the sub-characteristics under these major domains. I would like to know the origin and validity of this framework.

Response: Thank you for your insightful feedback. The framework of Figure 2 was specifically developed for this study. We built the framework based on the main challenges in the application of neuroprosthesis. Based on the literature review [1-4], the primary concerns encompassing prosthetic rehabilitation for upper limb amputations mainly involve the therapy efficacy, clinical applicability, rehabilitation motivation, and costs. Accordingly, this review is to show great promise of virtual technology in solving these issues of prosthetic rehabilitation applications. Therefore, we summarized the technology of virtual rehabilitation training from the four aspects:

Usability: Virtual approaches provide an intuitive training environment to improve motor functions step-by-step and alleviate the ceiling effect of physical rehabilitation.

Flexibility: Virtual approaches provide an extensible and flexible platform for multi-technology interaction.

Affinity: Virtual approaches enhance adherence and motivation for users to persist in long-term training.

Affordability: Virtual environments offer a low-cost and adjustable platform to enhance the skill proficiency of upper limb amputees.

Reference:
1. Biddiss, E.A.; Chau, T.T. Upper Limb Prosthesis Use and Abandonment: A Survey of the Last 25 Years. Prosthetics & Orthotics International 2007, 31, 236–257, doi:10.1080/03093640600994581.

2. Biddiss, E.; Chau, T. The Roles of Predisposing Characteristics, Established Need, and Enabling Resources on Upper Extremity Prosthesis Use and Abandonment. Disability and Rehabilitation: Assistive Technology2007, 2, 71–84, doi:10.1080/17483100601138959.

3. Biddiss, E.; Chau, T. Upper-Limb Prosthetics: Critical Factors in Device Abandonment. American Journal of Physical Medicine & Rehabilitation2007, 86, 977–987, doi:10.1097/PHM.0b013e3181587f6c.

4. Melton, D.H. Physiatrist Perspective on Upper-Limb Prosthetic Options: Using Practice Guidelines to Promote Patient Education in the Selection and the Prescription Process. J Prosthet Orthot2017, 29, P40–P44, doi:10.1097/JPO.0000000000000157.

We have revised the content to be more appropriate, as copied below. (Highlighted in Lines 471-475, in “discussion” Section):

The above four key points respectively encompasses the the primary concerns of prosthetic rehabilitation for upper limb amputation: therapy efficacy, clinical applicability, rehabilitation motivation and costs. Accordingly, this review is to show great promise of virtual technology in solving these issues of prosthetic rehabilitation applications.

Reviewer 3 Report

I have to say that the new manuscript is totally different from the first one and the authors have done a great work now to detail the findings to the required level of a review paper. 

Based on the new level of the paper and the answers of the authors, from my personal point of view the paper can be now published. 

Author Response

 Reviewer #3:

I have to say that the new manuscript is totally different from the first one and the authors have done a great work now to detail the findings to the required level of a review paper.

Based on the new level of the paper and the answers of the authors, from my personal point of view the paper can be now published.  

Response: Thanks for your comments. We express our gratitude for your patience and professionalism.